# Skeletonization of Lagrangian Point Clouds:
# Extracting Transport Networks from Particle Tracking Data

Viktor Škultéty[1,3]    Fabrice Guibert[1,3]    Nicolas Zucker[2]
Thomas Deffieux[2]    Mickael Tanter[2]    Dimitri Van De Ville[1,3]

[1]Neuro-X Institute, École Polytechnique Fédérale de Lausanne, Switzerland
[2]Physics for Medicine Paris, Inserm, ESPCI Paris, France
[3]Department of Radiology and Medical Informatics, University of Geneva, Switzerland

{viktor.skultety, fabrice.guibert, dimitri.vandeville}@epfl.ch
{nicolas.zucker, thomas.deffieux, mickael.tanter}@espci.fr

## Abstract

*Recovering transport pathways from particle-tracking measurements of fluid flows remains challenging due to the large number of detections, directional variability, and the lack of well-defined geometric boundaries. We propose a new algorithm based on a projected $L_1$ geometric median: points are attracted toward local centers of mass while motion along the local mean flow direction is suppressed. This constraint prevents collapse to density maxima without explicit regularization and operates directly on point coordinates, producing skeletons that are not constrained by voxel-grid discretization. Convergence is robust to random initialization given adequate sampling density, and the method remains scalable by operating on an aggregated representation of the data.*

*We demonstrate feasibility on simulated and experimental data. In particular, for in vivo 3D ultrasound localization microscopy (ULM), the method recovers the microvascular network of a rodent brain. The results indicate that flow-aware skeletonization provides a viable alternative to pipelines operating primarily on particle density representations.*

## 1. Introduction

Curve skeletons provide a simplified, compact representation of 3D data, capturing the essential structure. It is widely used in segmentation, registration, and shape analysis [4, 21]. In computer graphics and vision, extensive research has addressed skeleton extraction directly from point clouds, avoiding discretization or mesh reconstruction [6, 12, 14]. Most existing methods, however, assume that points sample an object surface, as is typical for sensing modalities such as LiDAR, RGB-D cameras, or multi-view photogrammetry [18]. Under this assumption, the skeleton corresponds to a medial structure of an underlying solid shape.

Lagrangian point clouds are a fundamentally different type of data produced by recent imaging modalities [20]. In flow measurement techniques such as particle tracking velocimetry, tracer particles are observed as they move with the flow [15]. These measurements enable reconstruction of complex 3D flow fields and have been used to study transport processes in systems such as porous media [5]. As a result, the data encode transport pathways through particle motion rather than surface geometry, producing samples concentrated along flow trajectories instead of object boundaries. A similar type of data arises in medical imaging: ultrasound localization microscopy (ULM) reconstructs microvasculature by tracking intravenously injected microbubbles [8, 9]. These detections are commonly rasterized onto a 3D grid, but the high spatial resolution leads to representations containing up to $10^9$ voxels, making conventional analysis computationally demanding and introducing discretization artifacts associated with the grid.

Therefore, recovering a skeleton representation of such data is important because it provides a compact description of the underlying transport network [2]. Instead of analyzing millions of individual particle detections, the flow structure can be represented as a graph of centreline curves encapsulating the connectivity and topology of transport pathways. Such representations enable quantitative analysis of transport networks, flow routing, and structural connectivity in complex biological and physical systems [19, 22].

Applying conventional point-cloud skeletonisation algorithms directly to Lagrangian point clouds is problematic because no surface is observed. Related approaches in data analysis extract curve-like structures as density ridges of volumetric point clouds, but rely solely on scalar den-

sity and therefore do not exploit the directional information present in flow measurements [7, 11, 17]. Trajectory-based methods, on the other hand, operate at the level of individual paths and are not formulated as a volumetric centerline estimation [1]. These limitations motivate a formulation tailored to densely sampled volumetric flow data.

In this work, we introduce *Lagrangian Point Cloud Skeletonization* (LPCS), a method for skeleton extraction from volumetric particle-tracking data. The approach exploits the local flow orientation of particle detections to recover the topology of transport pathways directly in continuous space. By attracting points toward local centers of mass in the plane orthogonal to the flow, the method recovers centerlines aligned with the underlying network while avoiding explicit estimation of geometric features. We demonstrate the approach on simulated datasets and on in-vivo 3D ULM measurements of rodent brain microvasculature.

The main contributions of this work are:

- A skeletonization method for volumetric Lagrangian point clouds, where geometry is inferred from oriented particle detections rather than surface samples.
- A projected $L_1$-type optimization algorithm that uses local flow direction to recover centerline structures while preventing collapse to density maxima without explicit regularization.
- A practical pipeline for large volumetric datasets, demonstrated on simulated data and on in-vivo ULM measurements of rodent brain microvasculature.

## 2. Related work

Skeleton recovery is essentially the retrieval of a curve that best summarizes the data. However, the data acquisition and representation play a major role in the method design.

### 2.1. Skeletons from Rasterized Images

Skeletonization of flow-derived volumetric images is common in biomedical imaging, where curve networks represent transport pathways such as vascular flow [16]. These methods typically operate on scalar representations of the measurements (e.g., intensity, contrast concentration, or velocity magnitude) rather than the vector flow. Tubular structures are commonly enhanced using multiscale vesselness filters based on the Hessian eigenstructure [10, 13], and centerlines are then extracted using distance transforms or minimal path techniques [2]. Such pipelines rely on dense voxel grids and therefore incur substantial computational cost and discretization artifacts tied to grid resolution.

### 2.2. Skeletons from Geometric Point Clouds

For point clouds, methods evolved from geometric contraction approaches that collapse the point set while preserving topology [6], to robust medial formulations estimating

interior symmetric points from local neighborhoods [12]. Recent work further improves robustness to incomplete data by exploiting additional geometric relations between points, for example through visibility analysis [23], and learning-based approaches infer skeletal structures directly from point sets [14].

Collectively, these methods treat the skeleton as a medial structure of an underlying solid object [21]. Consequently, they assume the points sample the object boundary, so that an interior–exterior separation is defined. For volumetric detections filling the interior of a tube, this assumption no longer holds: the available signal is dominated by point density rather than boundary geometry, making the task closer to density-ridge extraction than medial-axis recovery.

### 2.3. Density Ridge Estimation

A related line of work extracts structures as ridges of a density estimated from point samples. These ridges capture the principal modes of the distribution and have been used to recover skeleton-like representations in sampled point sets [7, 11]. They are typically computed by iteratively moving points along directions given by the local Hessian of the density [17]. While effective, the required repeated local eigendecompositions become computationally demanding for large point clouds.

In Lagrangian flow measurements, particle detections additionally carry directional information: nearby particles tend to follow locally consistent motion aligned with the underlying transport pathways [5, 15]. The structure of interest can therefore be characterized through the local coherence of flow orientation rather than solely through spatial density, motivating our formulation in which centerline extraction is constrained by the local flow direction.

### 2.4. Trajectory-Based Skeletonization

Beyond geometric and density-based approaches, skeletonization has also been studied in the context of trajectory analysis. In transportation research, trajectory-based map inference methods reconstruct road center lines directly from collections of GPS tracks [3]. A representative example is RoadRunner, which starts from an initial graph and iteratively refines it by following the flow of GPS trajectories to construct a road network [1]. Its optimization therefore depends on trajectory geometry and the quality of the initial graph, and it is not formulated as volumetric centerline extraction. These approaches are tailored to sparsely sampled vehicle-tracking data and operate at the level of individual trajectories rather than spatially aggregated, densely overlapping volumetric samples.

## 3. Overview

The input is an unevenly distributed set of oriented points $Q = \{(q_j, p_j)\}_{j \in 1 \dots J} \subset \mathbb{R}^3 \times \mathbb{R}^3$, and the output is a set of

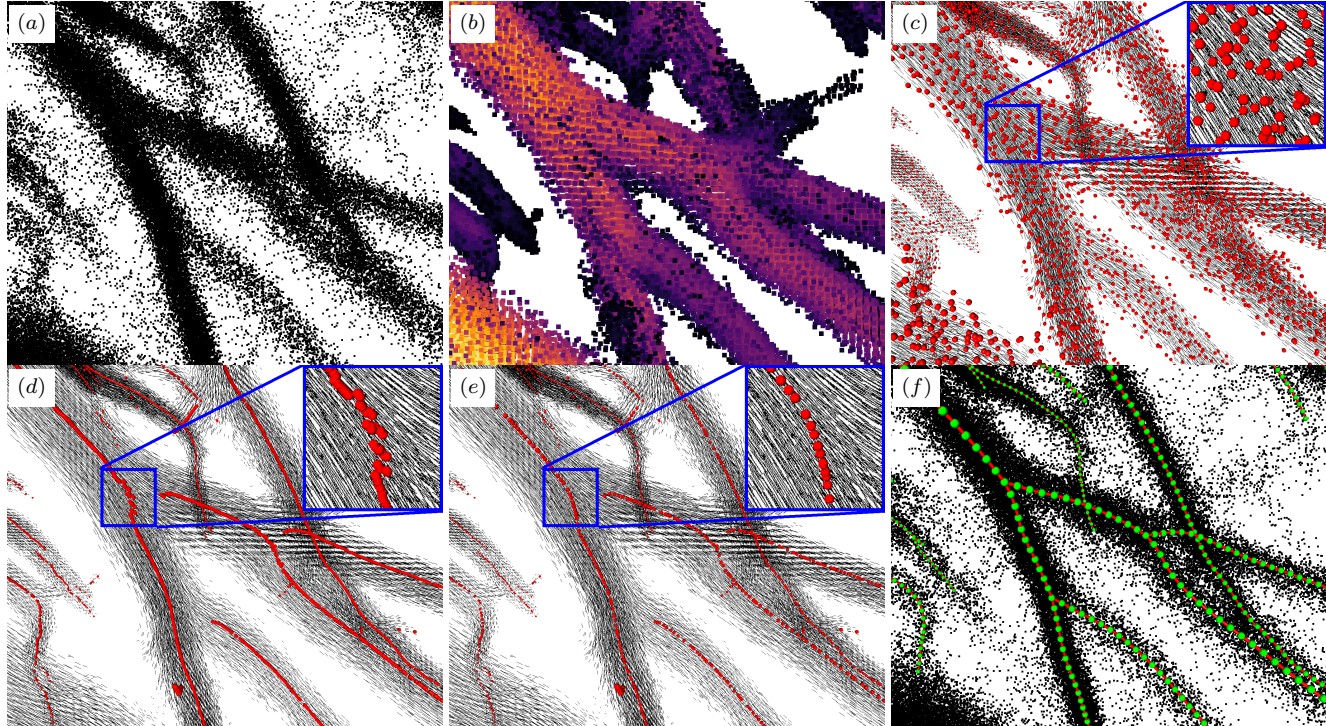

Figure 1. Overview of the LPCS algorithm. Starting from a dense Lagrangian point cloud, samples are attracted toward local centerlines using a projected optimization and subsequently refined to produce a connected skeleton graph. ($a$) Full Lagrangian point cloud (positions only). ($b$) Downsampled data with point mass encoded by colour. ($c$) Initial optimization samples (red points); dashed segments indicate local flow orientation. ($d$) Projected optimization attracting samples toward local centerlines. ($e$) Local refinement through a second optimization pass. ($f$) Final skeleton graph after smoothing and resampling.

oriented skeleton points $X = \{(x_i, n_i)\}_{i \in 1 \ldots I} \subset \mathbb{R}^3 \times \mathbb{R}^3$. The LPCS algorithm proceeds as follows; the individual steps are illustrated in Fig. 1:

1) Downsample $Q$ at a scale comparable to the minimum flow-channel radius and assign each retained point a mass (density weight) and a mean orientation; the retained points serve as attractors defining local centers of mass.

2) Initialize samples with approximately uniform spatial coverage and iteratively project them toward the local center of mass defined by nearby attractors within a prescribed radius, restricting displacement to the plane orthogonal to the local orientation.

3) In regions where the samples do not collapse to a one-dimensional structure, perform a second pass where the converged samples become the new attractors and the projection is repeated locally.

4) Extract and connect branches to form a skeleton graph, then smooth and resample to the desired spatial resolution.

## 4. Method

### 4.1. Projected $L_1$ Medial Skeleton

We seek skeleton points located at the center of local tubular structures. For very dense point clouds, operating directly on individual detections becomes computationally prohibitive. The point cloud is therefore downsampled by partitioning space into spatial bins at a resolution comparable to the smallest recoverable skeleton scale, such that the smallest vessels are represented by multiple bins across their diameter. Each bin is represented by the centroid of its detections, along with the corresponding mass and average orientation. This aggregation substantially reduces computational cost while avoiding discretization artifacts associated with rasterization, as subsequent processing operates on centroid positions rather than a discretized image grid.

After downsampling, we obtain a set of weighted oriented points $Q_m = \{(q_j, p_j, m_j)\}$. Residual noise is reduced by smoothing mass and orientation with a Gaussian kernel, whose full-width-at-half-maximum (FWHM) is chosen small enough to preserve small structural features. This provides stable local estimates for the subsequent optimization.

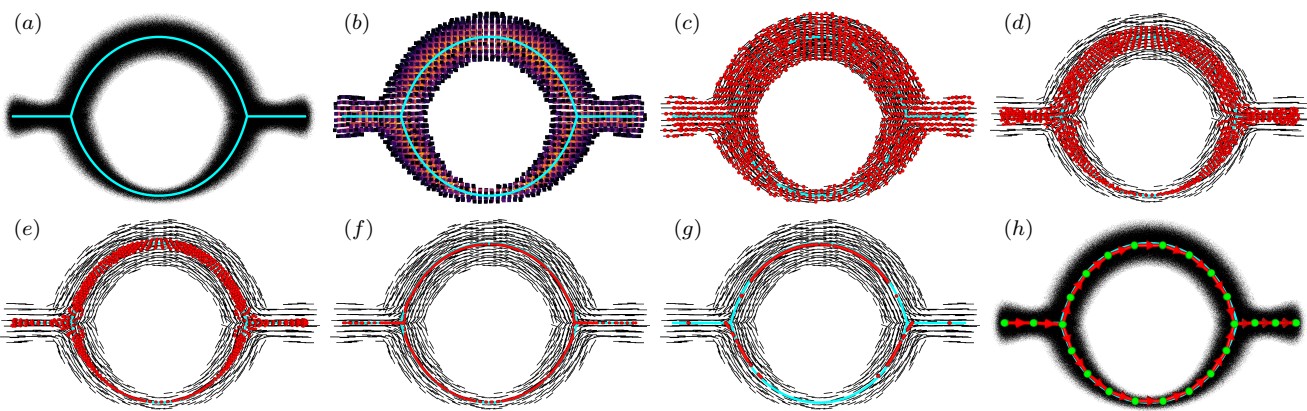

Figure 2. Synthetic example illustrating the LPCS pipeline and the role of the projected update. Cyan denotes the ground-truth centerline, and dashed segments indicate the local flow orientation. ($a$) Full Lagrangian point cloud (positions only). ($b$) Downsampled data with point mass encoded by color. ($c$) Initialization of the optimization: red points denote initial samples. ($d$) Early iteration of the projected update. ($e$) Later iteration of the projected update. ($f$) Final optimization, producing skeleton points aligned with the centerline. ($g$) Final optimization without projected update: points collapse toward density maxima and fail to form a curve skeleton. ($h$) Skeleton graph constructed from the converged points in ($f$).

We then define a loss function whose minimizer yields the skeleton points

$$\mathcal{L}(x_i) = \sum_j m_j \, \|x_i - q_j\| \, \theta_h(\|x_i - q_j\|), \qquad (1)$$

where $\theta_h(r) = \exp\!\left(-4r^2/h^2\right)$ is a fast decaying smoothing function defining the support of the minimizer. The parameter $h$ controls the spatial scale of interaction and is chosen on the order of the characteristic radius, balancing convergence stability and accurate centerline localization. The objective corresponds to a local geometric median, attracting the point toward the center of nearby samples.

Minimization of this energy leads to the Weiszfeld iteration

$$x_i^{k+1} = \frac{\sum_j q_j m_j \alpha_{ij}^k}{\sum_j m_j \alpha_{ij}^k}, \quad \alpha_{ij}^k = \frac{\theta_h(\|x_i^k - q_j\|)}{\|x_i^k - q_j\|}. \qquad (2)$$

Direct optimization of (2) collapses points to density modes. To recover a curve rather than a set of isolated clusters, we constrain motion to directions orthogonal to the local flow orientation. The update is therefore projected

$$\tilde{x}_i^{k+1} = x_i^k + \left(x_i^{k+1} - x_i^k\right) \cdot \mathbb{P}(n_i), \qquad (3)$$

where $\mathbb{P}(n_i) = I - n_i n_i^\top$ is the projection operator and the local orientation is defined as

$$n_i = \frac{\sum_j p_j' \, \theta_h(\|x_i^k - q_j\|)}{\left\|\sum_j p_j' \, \theta_h(\|x_i^k - q_j\|)\right\|}. \qquad (4)$$

The projection removes motion along the flow direction, allowing convergence only in transverse directions and

thereby recovering a 1D structure. When the converged points do not form a locally 1D structure, a refinement stage is applied in which the converged points become new attractors and the optimization is repeated locally. The resulting procedure defines the LPCS algorithm.

Figure 2 illustrates the algorithm on synthetic data with simulated attenuation. Without the constrained update, points collapse toward local density maxima, leaving sparsely sampled regions uncovered. The projected update prevents this collapse by restricting optimization to directions perpendicular to the flow. In this synthetic example the refinement stage is not activated, as the points already converge to a 1D structure. The refinement step is illustrated later for experimental data in Fig. 5.

## 4.2. Graph Construction

To obtain a structured representation of the skeleton, we first downsample the set of skeleton points to reduce redundancy and simplify subsequent processing. A relative neighborhood graph is then constructed, restricted to a maximum connection distance in order to prevent linking spatially distant branches. The adjacency matrix of this graph is used to identify individual branches, which are subsequently smoothed and resampled. Branches shorter than a user-defined length threshold are discarded as spurious fragments. Finally, the directionality of each skeleton point is inferred from the local flow. The skeleton graph of a synthetic dataset is shown is shown in Fig. 2($h$).

Even after construction of the relative neighborhood graph, certain regions may remain disconnected; see Fig. 3. This behavior is expected; In physical flows, trajectories entering a daughter branch typically originate from the bound-

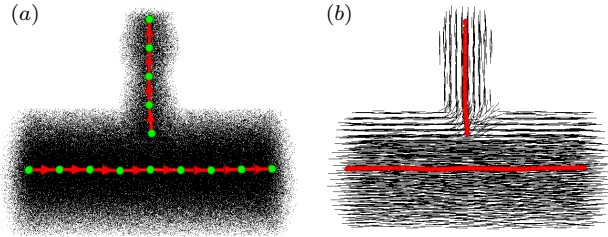

(a)       (b)

Figure 3. Disconnected skeleton segments caused by flow topology. (a) Point cloud with the final skeleton graph. (b) Streamlines and contracted skeleton points before resampling. The apparent gap is not an algorithmic failure but a consequence of the transport structure: trajectories entering the daughter branch originate near the boundary of the parent channel, so no centerline passes continuously through the junction.

ary of the parent branch rather than its centerline, and therefore the centerline points do not provide a direct connection. Hence, the resulting discontinuity reflects the structure of the flow rather than a limitation of the algorithm.

Whether explicit reconnection is required depends on the downstream application. Tasks involving full graph analysis may require a connected topology, whereas individual branch flow analysis pipelines might not. When needed, connections can be introduced by identifying terminal nodes of the relative neighborhood graph and linking them to the nearest point belonging to a neighboring branch.

## 5. Experiments

### 5.1. Dataset and Processing Pipeline

We evaluate the method on a dataset obtained from 3D ULM imaging of rat brain microvasculature [8, 9]. The dataset and source code are proprietary and cannot be publicly released due to intellectual property constraints. We provide detailed descriptions of their characteristics and the evaluation protocol to ensure transparency.

The data were acquired from an anesthetized rat following craniotomy between Bregma and Lambda. ULM imaging was performed over a 40-minute period with continuous injection of microbubble contrast agent. The final dataset consists of a volume of size $5 \times 10 \times 10 \text{mm}^3$ containing approximately $2 \times 10^8$ localized contrast agent detections, representing sparse and noisy samples of the underlying vascular network. Our goal is to recover a skeleton of vessels with radii on the order of $\Delta = 10 \mu\text{m}$, which defines the aggregation scale used throughout the pipeline. The overall processing pipeline is illustrated in Fig. 1, and the main steps are described below.

The point cloud is first downsampled to a spatial resolution of $\Delta$, after which local masses and velocities are computed. Velocities are smoothed using a Gaussian kernel (FWHM = $4\Delta$). Points with low local support ($m_j < 3$)

are removed, yielding an average reference-point density of $3 \times 10^4 \text{ mm}^{-3}$, while local densities remain strongly non-uniform and can reach up to $10^6 \text{ mm}^{-3}$ in large vessels. To improve convergence, a nonlinear feature lift is applied to the mass values, $m_j \rightarrow \exp(m_j)$. To obtain uniform coverage of the domain, optimization samples are initialized using Poisson-disk sampling with a minimum separation of $2\Delta$ between samples. This prevents oversampling in densely populated regions while maintaining adequate coverage in sparse areas.

The LPCS algorithm is then applied in three stages with increasing interaction radius:
1) 50 iterations with $h = 3\Delta$.
2) 100 iterations with $h = 4\Delta$.
3) 150 iterations with $h = 5\Delta$.
This multiscale schedule improves convergence while preserving structural detail; the effect of the interaction radius is illustrated in Fig. 4. Points moving less than $\Delta/100$ between iterations are considered converged and are not updated in subsequent iterations or stages.

Real data contain measurement noise and other artifacts not present in a synthetic setting; as a result, some branching regions do not collapse to a well-defined curve (Fig. 5). To address this, we perform a local refinement:
1) Downsample the obtained skeleton points to $\Delta$.
2) For each sampled point, compute PCA of the neighboring skeleton points within a $4\Delta$ neighborhood and obtain eigenvalues $\lambda_1 \geq \lambda_2 \geq \lambda_3$. Points with $\lambda_3/\lambda_1 > 10^{-4}$ are classified as non-curve points.
3) Re-run the skeletonization algorithm in these regions using the current skeleton points as reference points.
The resulting set of points is again downsampled to $\Delta$ and converted into a graph using relative neighborhood graph with distance threshold $3\Delta$. Individual branches are identified form graph adjacency matrix, and are further smoothed and resampled to $2\Delta$ arc length. Branches shorter than $10\Delta$ are discarded. Finally, branch endpoints are detected and connected to the nearest point on a neighboring branch within a spatial radius of $10\Delta$. The connection is accepted only if the angle between the local flow direction at the endpoint location and the displacement toward the candidate point is less than $20°$.

### 5.2. Results

The extracted skeleton reveals a dense microvascular network spanning the imaged volume (Fig. 6). The resulting graph consists of numerous interconnected branches representing the vascular tree across the entire field of view, with many bifurcations reflecting its hierarchical structure. The reconstruction captures vessels of varying sizes present in the data.

Poisson-disk initialization provides uniform spatial coverage while preventing oversampling in densely populated

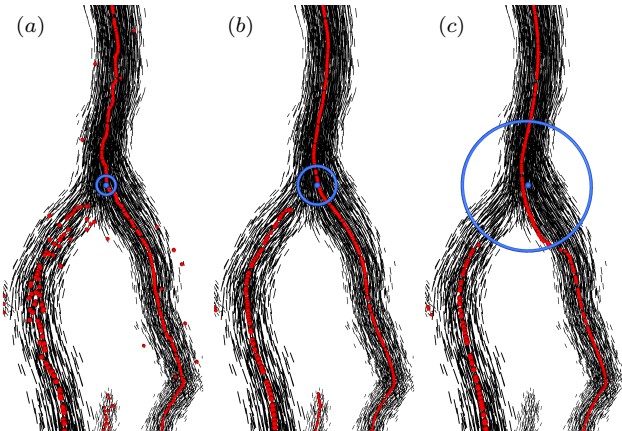

Figure 4. Effect of the attraction radius $h$ during the initial optimization stage (before refinement). $(a)$ When $h$ is too small, the attraction is too local, causing slow or incomplete convergence to the centerline. $(b)$ An intermediate value yields stable convergence and accurate reconstruction. $(c)$ Large $h$ oversmooths the geometry and fails to resolve fine details. The blue circle shows the local neighborhood defined by $h$.

regions, enabling recovery of vessels across a wide range of scales. The method is robust to random initialization provided that the sampling density is sufficiently high. Although some regions do not initially collapse to a locally one-dimensional curve, the refinement step promotes convergence to a curve-like skeleton in these areas. Discontinuities at bifurcations arising from the flow topology are subsequently corrected during graph construction.

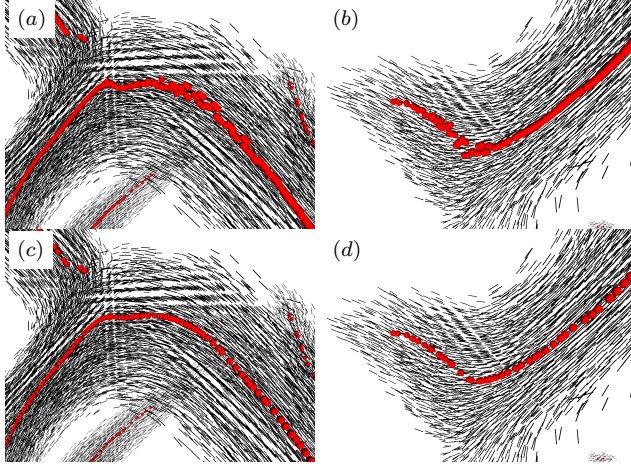

Figure 5. Local refinement following the initial optimization. $(a)$ Turning region where the optimization fails to collapse to a one-dimensional curve. $(b)$ Measurement artifact where the flow abruptly terminates. $(c)$ Result after refinement in the turning region. $(d)$ Result after refinement in the measurement artifact region. Black dashes: local flow; red: skeleton points.

## 5.3. Limitations

The method depends on the density of the initial optimization samples. Although robust to random initialization, insufficient sampling density may leave some regions undersampled and prevent accurate recovery of the skeleton. The contraction stage also depends on the kernel support parameter $h$, which defines the region of attraction. Large values may merge nearby vessels, while small values may fail to capture larger structures. In our implementation this trade-off is mitigated by applying the optimization in multiple stages with increasing interaction radii.

The nonlinear feature lift applied to the local mass improves convergence by emphasizing strongly supported structures. However, because it is based on an exponential transformation, it can produce extremely large values when the aggregated mass becomes high, particularly at coarse downsampling resolutions. In such cases, care must be taken to avoid excessive dynamic range. Possible remedies include normalizing the mass by a reference value before exponentiation, or replacing the exponential lift with a milder power-law transformation.

The final skeleton graph is constructed using a relative neighborhood graph, which requires selecting a distance threshold for connecting points. If the threshold is too small the graph may become disconnected, whereas overly large values may link unrelated vessels. In addition, discontinuities may arise near bifurcations due to the topology of the underlying flow field. In the current implementation these are resolved using a simple proximity-based and orientation-based connection rule, which may not always perfectly recover the underlying centerline geometry.

## 5.4. Comparison with Prior Work

Biomedical skeletonization pipelines typically operate on rasterized volumetric data. While effective for many imaging modalities, voxelization introduces discretization artifacts. The proposed approach instead operates directly on point coordinates, producing a continuous centerline representation that is not constrained by voxel-grid discretization. This difference is illustrated in Fig. 7.

Many point-cloud skeletonization methods assume that points sample the surface of an object and recover the medial structure of the enclosed volume. In Lagrangian flow measurements, however, detections occupy the interior of transport pathways rather than their boundaries, making such formulations unsuitable. Density ridge estimation provides a closer connection, as it extracts curve-like structures from volumetric point distributions. Classical ridge algorithms infer ridge directions from the Hessian of a density field and therefore require repeated eigendecomposition during optimization. In contrast, the proposed method estimates ridge directions directly from the local flow orientation. Furthermore, the high redundancy of Lagrangian de-

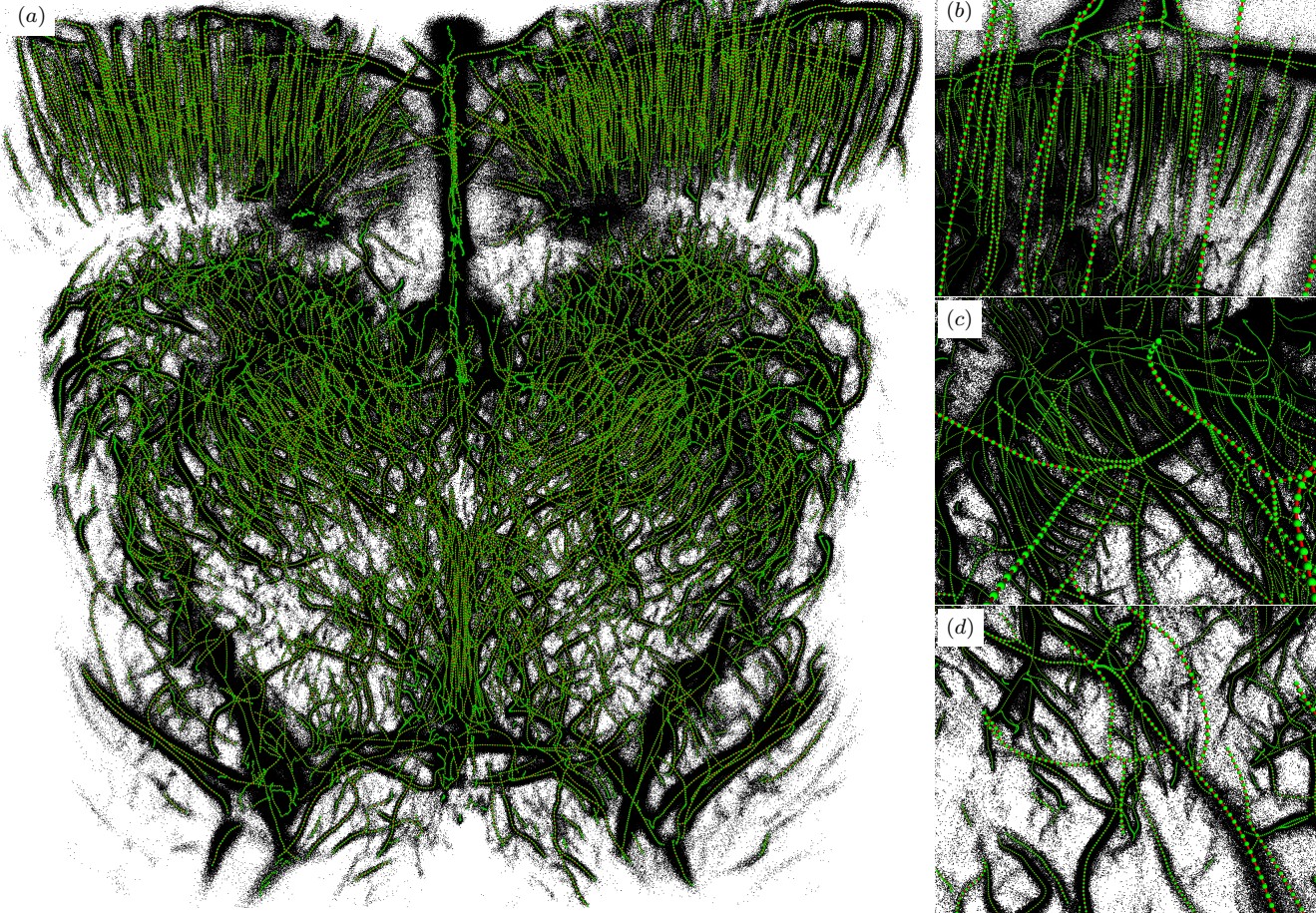

Figure 6. Skeleton reconstruction from experimental 3D ULM data using LPCS. $(a)$ Full vascular skeleton extracted from brain-wide microvasculature, shown with centerline sampling of $35\,\mu$m. The reconstructed graph reveals a dense network of interconnected branches spanning the imaged volume. $(b)$–$(d)$ Enlarged views of local regions displayed with finer $20\,\mu$m sampling, highlighting small vessels and complex bifurcations. Local sampling density provides spatial reference for the 3D view.

tections allows spatial aggregation before optimization, so the computational cost scales with the number of reference points rather than the total number of detections.

Trajectory-based network reconstruction methods operate directly on individual paths rather than volumetric samples. For example, RoadRunner iteratively constructs a road network by following the flow of GPS trajectories starting from an initial graph. Similar to tractography, such methods recover network structure by tracing trajectories, and the resulting paths are therefore sensitive to the initialization. In contrast, the proposed formulation treats the detections as samples of a volumetric transport field and estimates the centerline from their collective spatial and directional structure. Because optimization samples are initialized randomly and iteratively attracted toward local centers of mass, the method converges directly to the centerline of the transport pathways.

## 6. Conclusion

We introduced *Lagrangian Point Cloud Skeletonization*, a framework for extracting transport centerlines from volumetric Lagrangian point clouds. Optimization is restricted to directions orthogonal to the flow, preventing collapse to density maxima, and the resulting skeleton curves are not constrained by voxel-grid discretization. The method was demonstrated on simulated and in-vivo 3D ULM data, where it successfully recovered the microvascular network of a rodent brain. These results show that incorporating flow orientation provides an effective way to extract transport structures from volumetric particle detections.

Future work could focus on improving reconstruction near complex bifurcations and developing more robust graph construction strategies. Automatic selection of optimization parameters would further improve robustness across datasets. The framework could also be extended to

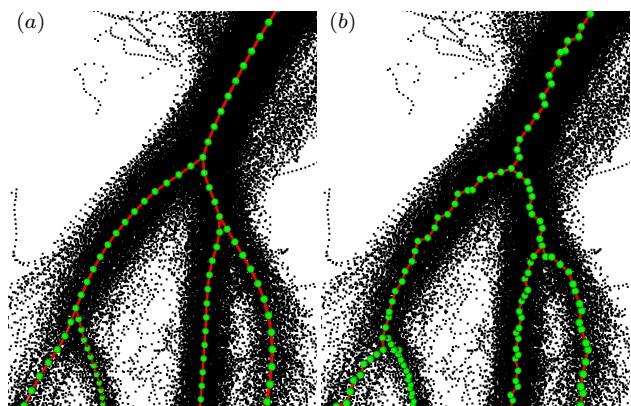

Figure 7. Comparison with a standard rasterized skeletonization pipeline used for ULM data. (*a*) LPCS: skeleton points extracted directly in continuous space. (*b*) Skeleton obtained from a rasterized vesselness-based approach [2, 13] at a voxel resolution of $\Delta$ and mapped back to point coordinates. Discretization introduces grid artifacts that affect the geometric accuracy of the centerline.

rasterized vector data by converting non-zero voxels into weighted point samples, enabling sub-voxel skeleton extraction.

## Acknowledgments

This research was supported in parts by the Swiss State Secretariat for Education, Research and Innovation (SERI) under contract number 22.00161, and by the EIC Pathfinder MICROVASC.

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
