# OpenReview forum: "Skeletonization of Lagrangian Point Clouds: Extracting Transport Networks from Particle Tracking Data"
_thecvf.com/CVPR/2026/Workshop/3D4S — CVPR 2026 Workshop 3D4S Poster_

### Official Review · Reviewer_5Cmc · 2026-04-20
**A novel and well-motivated method for flow-aware skeletonization of Lagrangian point clouds, with compelling visual results**

**Rating:** 7
**Confidence:** 3

**Review:**

This paper introduces Lagrangian Point Cloud Skeletonization (LPCS), a method for extracting transport network centerlines from volumetric particle-tracking data. The key idea is a projected L1 geometric median: points are attracted toward local centers of mass, but motion along the local flow direction is suppressed via orthogonal projection, so they converge to centerlines rather than collapsing to density maxima. The formulation is clean, well-motivated, and addresses a genuine gap as existing skeletonization methods assume surface sampling or ignore directional information, while Lagrangian point clouds fill tube interiors and carry velocity vectors. The paper demonstrates the method on synthetic data and on in-vivo 3D ultrasound localization microscopy (ULM) of rodent brain microvasculature, with visually compelling results. The dataset and source code are proprietary, which limits reproducibility. Citation integrity is strong.

---

### Official Review · Reviewer_G7ZD · 2026-04-22
**Caims are plausible but insufficiently validated.**

**Rating:** 5
**Confidence:** 3

**Review:**

The topic of this paper is niche but meaningful. The idea is novel, and the paper shows good visualization results. However, the weaknesses are also obvious. My main concerns are as follows:

The results are not quantitatively evaluated. No numerical metrics are provided to demonstrate the effectiveness of the method. At least for the ablation study, quantitative results are important. In addition, no runtime, memory usage, or complexity analysis is provided.
The pipeline contains many parameters across different steps. How are these parameters selected? Are they specifically tuned for this dataset?
If the dataset will not be released, it will be difficult for others to follow the work or reproduce the results.

---

### Official Review · Reviewer_C5wU · 2026-04-24
**This paper introduces Lagrangian Point Cloud Skeletonization (LPCS), a novel algorithm to extract continuous 1D transport networks from dense, volumetric particle-tracking data. By constraining a projected L1 geometric median optimization to the plane orthogonal to local flow orientations, the method effectively mitigates density collapse and avoids voxel-grid discretization artifacts.**

**Rating:** 7
**Confidence:** 4

**Review:**

Quality: The theoretical foundation of the algorithm is elegant and well-reasoned. The use of the projection operator $P(n_i) = I - n_in_i^\top$ to restrict the L1 median update is a highly effective way to leverage the vector nature of Lagrangian data. However, the experimental validation is lacking in rigor. While the visualizations of the ULM data are impressive, there is a distinct absence of quantitative evaluation. There are no numerical comparisons against competing methods (e.g., rasterized Hessian-based vesselness filters or density ridge estimators) on synthetic datasets where ground-truth topology and centerlines are known.Clarity: The paper is well-structured and clearly written. The motivation is excellent; the authors do a fantastic job explaining exactly why Lagrangian point clouds require a fundamentally different treatment than traditional LiDAR or RGB-D geometric point clouds. The methodology is easy to follow, and the limitations are discussed candidly.
Originality: The work exhibits strong originality. While $L_1$-medians for skeletonization have been explored in computer graphics (e.g., $L_1$-medial skeleton), adapting this framework to volume-filling flow data by tying the spatial update constraint directly to the localized velocity vectors is a novel and clever contribution.
Significance: This approach has high potential significance for biomedical imaging (specifically ULM) and fluid dynamics (particle tracking velocimetry). Bypassing the heavy memory constraints and quantization artifacts of massive dense voxel grids allows for much more scalable and geometrically accurate analyses of transport networks.

Pros

Domain-Aware Formulation: Integrating directional flow data directly into the spatial optimization step to prevent density collapse is a highly principled approach to this specific data modality.

Continuous Domain Processing: By avoiding intermediate rasterization, the method bypasses resolution-tied discretization artifacts, producing smooth sub-voxel representations.

Scalability: The initial aggregation of the raw point cloud into localized mass/orientation bins, coupled with Poisson-disk initialization, ensures the algorithm remains computationally tractable despite the massive size of typical ULM datasets.

Cons:
Lack of Quantitative Benchmarking: The evaluation is purely qualitative. The authors introduce a synthetic dataset in Figure 2 to illustrate the pipeline, but they do not use it to report quantitative metrics (e.g., topological error, Chamfer distance to ground truth, bifurcation detection accuracy) against standard baselines.Fragile Heuristics in Graph Construction: The post-processing steps for reconnecting the graph at bifurcations rely on hardcoded thresholds (e.g., a spatial radius of $10\Delta$ and a 20-degree angle limit). As noted in the paper, flow topology naturally causes gaps at bifurcations. Relying on simple proximity/angle heuristics to bridge these gaps seems brittle and likely to fail in complex, dense capillary beds.Numerical Instability of Mass Lift: The authors acknowledge that the exponential feature lift $m_j \rightarrow \exp(m_j)$ can produce extremely large values, causing dynamic range issues. A more stable formulation (e.g., a tuned power-law or normalized softmax) should be formally integrated into the method rather than just suggested as a future remedy.

---

### Decision · Program_Chairs · 2026-04-28

Accept (Poster)